



# Lipid biomarker-based sea (sub)surface temperature record offshore Tasmania over the last 23 million years

Suning Hou[1], Foteini Lamprou[1], Frida S. Hoem[1], Mohammad Rizky Nanda Hadju[1], Francesca Sangiorgi[1], Francien Peterse[1], Peter K. Bijl[1]

[1]Department of Earth Sciences, Utrecht University, Utrecht, 3584CB, the Netherlands

*Correspondence to*: Suning Hou (s.hou@uu.nl)

**Abstract:**

The Neogene (23.04–2.58 Ma) is characterized by progressive buildup of Antarctic and Northern Hemisphere ice volume and climate cooling. Heat/moisture delivery to Antarctica is to a large extent regulated by the strength of meridional temperature
gradients. However, the evolution of the Southern Ocean frontal systems remains scarcely studied in the Neogene. Here we present the first long-term continuous sea (sub)surface temperature (SST) record of the subtropical front area in the Southern Ocean at Ocean Drilling Program (ODP) Site 1168 off western Tasmania. This site is at present located near the subtropical front (STF), as it was during the Neogene, despite a 10 degree northward tectonic drift of Tasmania during the Neogene. We analyzed glycerol dialkyl glycerol tetraethers (GDGTs, on 433 samples) and alkenones (on 163 samples) and reconstructed the
paleotemperature evolution using $TEX_{86}$ and $U^{k'}_{37}$ as two independent quantitative proxies. Both proxies indicate that Site 1168 experienced a temperate ~25°C during early Miocene (23–17 Ma), reaching ~29 °C during the mid-Miocene Climatic Optimum. The stepwise ~10°C cooling (20–10°C) in the mid-to-late Miocene (12.5–5.0 Ma) is larger than observed in records from lower and higher latitudes. From the Pliocene to modern (5.3–0 Ma), STF SST first plateaus at ~15°C (3 Ma), then decreases to ~6°C (1.3 Ma), and eventually increases to the modern levels around ~16 °C (0 Ma), with a higher variability of
5 degrees compared to the Miocene. Our results imply that the latitudinal temperature gradient between the Pacific equator and STF during late Miocene cooling increased from 4°C to 14°C. Meanwhile, the SST gradient between the STF and the Antarctic margin decreased due to amplified STF cooling compared to the Antarctic Margin. This implies a narrowing SST gradient in the Neogene, with contraction of warm SSTs and northward expansion of subpolar conditions.



## 1. Introduction

Sea (sub)surface temperature (SST) reconstructions (Super et al., 2018, 2020; Sangiorgi et al., 2018; Tanner et al., 2020; Herbert et al., 2016; Zhang et al., 2014; Rousselle et al., 2013; Van der Weijst et al., 2022) and benthic foraminiferal oxygen isotopes (Westerhold et al., 2020; Holbourn et al., 2013; Lear et al., 2015; Lewis et al., 2007; Leutert et al., 2021) demonstrated that Neogene climate cooling occurred stepwise, with episodes of intermittent warming, e.g., at the mid-Miocene climatic optimum (MCO, 16.9–15 Ma) and mid-Pliocene warm period (mPWP, 3.264–3.025 Ma). This cooling trend is further accompanied by Antarctic ice volume increase (Lear et al., 2015; Lewis et al., 2007; Leutert et al., 2021), $pCO_2$ decline (Sosdian et al., 2018; Super et al., 2018; Tanner et al., 2020; Rae et al., 2021), strengthening of the Antarctic Circumpolar Current (ACC, Sijp et al., 2014; Evangelinos et al., 2021) and sea ice expansion (McKay et al., 2012; Sangiorgi et al., 2018; Bijl et al., 2018). The Southern Ocean is of special importance in reconstructions of past climate, as it plays a crucial role in ocean circulation, ocean-atmosphere carbon exchange, and as modulator of heat transport towards the largest body of land ice on Earth, the Antarctic ice sheet (Rintoul et al., 2018). The latitudinal position and strength of the ACC and its associated ocean fronts are forced by position shifts of the westerlies and bathymetry and have been suggested to modulate ocean-atmosphere $CO_2$ exchange as a feedback to the climate system (Toggweiler et al., 2006; Skinner et al., 2010). The gradual widening of Tasmanian gateway and Drake Passage in the Neogene provided the geographic boundary conditions for a further strengthening of the ACC and associated oceanic fronts (Sijp et al., 2014; Evangelinos et al., 2021). Yet, the evolution of the ACC in the Neogene is poorly documented, as its strength and position are difficult to constrain from geological archives. One of the manifestations of a strengthening ACC and frontal systems would be an increase in the meridional temperature gradient in the Southern Ocean, and the gradient between the Antarctic Margin and the subtropical front (STF) in particular. The latter represents the northern limit of the Southern Ocean, the northern branch of the ACC, and the boundary between the subtropical gyre and the subpolar waters, representing an oceanographic midpoint between the equator and Antarctica. While individual SST reconstructions for the Neogene Southern Ocean exist (e.g., Herbert et al., 2016), the evolution of the latitudinal SST gradient has thus far not been evaluated. Although a compilation of Antarctic ice-proximal SSTs has recently become available (Duncan et al., 2022), SST reconstructions from the more northern parts of the Southern Ocean only cover short time intervals, which precludes an integrated overview of the evolution of the latitudinal SST gradient.

We here provide a detailed reconstruction of the Neogene SST evolution of the subtropical front based on lipid biomarkers stored in sediments retrieved from Ocean Drilling Program (ODP) Site 1168, offshore western Tasmania. We base our reconstruction on two independent SST proxies. The TEX$_{86}$ paleothermometer is based on the relative number of cyclopentane



moieties in isoprenoid glycerol dialkyl glycerol tetraethers (isoGDGTs) produced by marine archaea, which varies as a function of ambient temperature in a global set of marine surface sediments (Schouten et al., 2002). The $U^{k'}_{37}$ index is based on the

relative abundance of di- and tri-unsaturated $C_{37}$ alkenones synthesized by unicellular haptophyte marine algae which yield a robust relationship with SST (Eglinton & Eglinton, 2008; Volkman et al., 1980). We put our new record into context of those from regions further north and closer to Antarctica for an integrated reconstruction of Southern Ocean latitudinal SST gradients.

## 2. Material and methods

### 2.1 ODP Site 1168

Site 1168 (42°36.5809′S; 144°24.7620′E; 2463 m modern water depth) (Fig. 1) is located on the continental slope of the west-Tasmanian continental margin. The site sits on the north edge of the Subtropical Convergence zone, which separates warm, saline subtropical waters from comparably cold and fresh subantarctic water masses (Exon et al., 2001; Heath et al., 1985). During the Neogene, the location of Site 1168 tectonically drifted along with Tasmania and Australia from 52°S at 23 Ma to its modern position at 42°S (Van Hinsbergen et al., 2015). During this northward tectonic drift, the Southern margin of Australia

was continuously bathed by the eastward flowing proto-Leeuwin Current (McGowran et al., 2004; Hoem et al., 2021). Hence, Site 1168 is well-suited to study the Neogene evolution of the ACC and the STF.



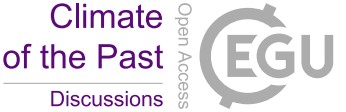

Figure 1: Neogene Paleogeographic maps of the Australian-Antarctic sector, with Deep Sea Drilling Program, the Ocean Drilling Program and Integrated Ocean Drilling Program site locations referred in this study. a, b, c: Reconstructed map of studied area using GPlates (Torsvik et al., 2012; Van Hinsbergen et al., 2015) with inferred surface ocean currents (red and blue solid lines, De Vleeschouwer et al., 2019; Jackson et al., 2019; Sauermilch et al., 2021). The edge of the light grey fill denotes present-day shorelines. The dark-grey contours indicate the edge of continental plates. Compiled sites and the site of this study are shown with black circles





**and red star respectively. d: Modern map (modified from NOAA) of studied area filled with modern sea surface temperature, which is indicated by colours and contours and numbers on the contours. The white line indicates the subtropical front.**

**2.2 Age model**

The post-cruise bio-magnetostratigraphic age model includes nannofossil, planktonic foraminifer, diatom, radiolarian, and dinocyst biostratigraphy with constraints from magnetostratigraphy and stable isotope data (Stickley et al. 2004). Here we recalibrated these datums to the Geological Time Scale 2020 (Gradstein et al., 2020), by using state-of-the-art biostratigraphic constraints from nannotax and foramtax, and updated diatom biostratigraphic constraints (Cody et al., 2008). We then fitted a

loess smooth curve through these datums whereby we assign a 10-fold weight on magnetostratigraphic and benthic $\delta^{18}O$ tie points. We interpolated this loess curve to obtain ages for the samples. We derive average sediment accumulation rates of 1.8 cm/kyr for the top 360 meters (22–0 Ma) and 7.1 cm/kyr between 360 and 462 meters below sea floor (mbsf; 23–22 Ma; Fig.2).

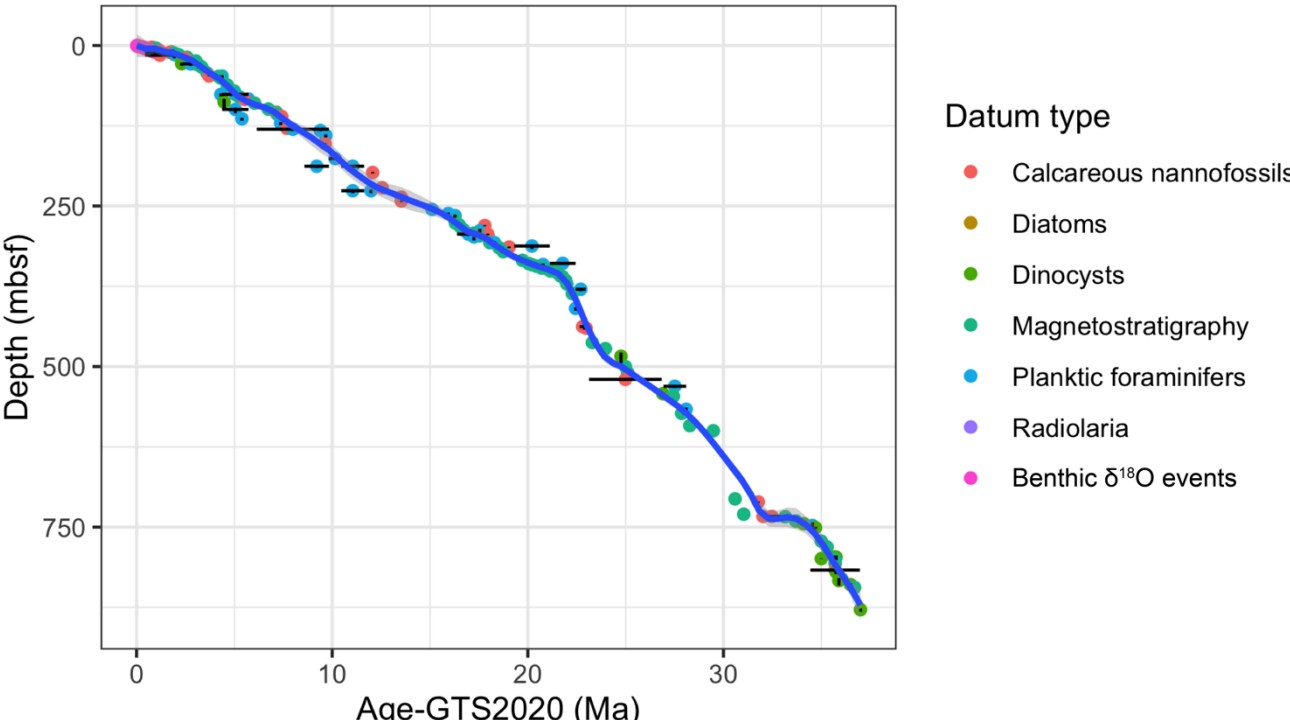

**Figure 2: Age model of Site 1168. Points indicate the datums (Stickley et al., 2004). Colors indicate datum types. Blue curve indicates**

**the loess smooth curve with a span of 0.1 throughout the studied interval, which we resampled to obtain ages for the samples used in this study.**



## 2.3 Lithology

A total of 883.5 meters of sediment was recovered from Site1168 Hole A, dating back to the Late Eocene to modern (Exon et al., 2001). The Neogene interval is represented in the upper 413 meters. Between 260–413 mbsf (early to mid- Miocene; 23.0–

15.6 Ma), sediments comprise of clay-bearing nannofossil chalk with a gradual decrease of non-carbonate minerals (Robert, 2004). The upper 260 meters (mid-Miocene to modern; 15.6–0 Ma) contain calcareous biogenic oozes, with 85–97 wt% calcium carbonate (Exon et al., 2001), a sharp decrease in detrital clay content occurs at the boundary between these lithologic units (Robert, 2004).

## 2.4 Biomarker extraction and analysis

Lipid biomarkers were extracted from 314 powdered and freeze-dried samples with a Milestone Ethos X microwave system using dichloromethane: methanol (DCM:MeOH) 9:1 (v/v). Activated $Al_2O_3$ columns were used for the separation of the total lipid extract into 3 fractions, using solvent mixtures hexane: DCM 9:1 (v/v), hexane: DCM 1:1 (v/v) and DCM: MeOH 1:1 (v/v) for apolar, ketone and polar fractions, respectively. Polar fractions were filtered using a 0.45 μm polytetrafluorethylene filter and analysed using an Agilent 1260 Infinity series HPLC system coupled to an Agilent 6130 single quadrupole mass

spectrometer, following instrumental and analytical setup as described in (Hopmans et al., 2016). 99ng of $C_{46}$ standard was added to the polar fraction in order to quantify the absolute concentration of GDGTs. The ketone fractions of 43 out of 314 samples were dissolved in ethyl acetate and analysed on a Gas Chromatograph (GC) coupled to a flame ionisation detector (GC-FID, Hewlett Packard 6890 series) equipped with a CP-Sil 5 fused silica capillary column (25 m x 0.32 mm; film thickness 0.12 um) and a 0.53 mm precolumn. Samples were injected on-column at 70°C with helium as a carrier gas and a flow rate of

2 ml/min. The oven program was as follows: 70°C for 1 min, then ramped to 130°C at 20°C /min, then to 320°C at 4°C/min, and then held isothermal for 10 mins. Di- and tri-unsaturated $C_{37}$ alkenones were identified based on retention time. Data is stored at Zenodo (Bijl et al., 2022).

## 2.5 SST reconstruction and confounding factor indices

We follow the approach by Sluijs et al. (2020) and Bijl et al. (2021) to assess non-temperature factors on the relative distribution

of isoGDGTs, and thus the $TEX_{86}$ value they represent. Briefly, this involves checking the weighted average of cyclopentane



moieties of isoGDGTs compared to modern values (with the Ring Index; Zhang et al., 2016), overprints from methanotrophic archaea (with the Methane Index; Zhang et al. 2011, Weijers et al., 2011), methanogens (with the GDGT-0/Cren ratio; Blaga et al., 2009), as well as contributions from deep-dwelling archaea (with the GDGT-2/GDGT-3 ratio; Taylor et al., 2013) or terrestrial GDGTs (with the BIT index; Weijers et al., 2006, Hopmans et al., 2004).The BIT index is determined by the ratio

of branched GDGTs (brGDGTs) produced by terrestrial bacteria and marine originated crenarchaeol. However, recent studies have proved that brGDGTs can be produced in situ in marine environments (Peterse et al., 2009; Sinninghe Damsté, 2016; Dearing Crampton-Flood et al., 2019). Thus, source(s) of brGDGTs are assessed using the weighted number of cyclopentane moieties in tetramethylated branched GDGTs ($\#rings_{tetra}$), where a value >0.7 is assumed to indicate a marine rather than a terrestrial source of these compounds (Sinninghe Damsté, 2016).

Numerous calibrations have been implemented to translate $TEX_{86}$ into sea surface temperature (e.g. Schouten et al., 2002; Kim et al., 2010; Tierney and Tingley, 2014). However, improved understanding of archaea ecology questions the validity of $TEX_{86}$ as a true proxy for the ocean mixed layer temperature. This is especially due to the variable export production zone depth (50-200m) of marine Thaumarchaeota. Fortunately, this can be revealed by the GDGT-2/GDGT-3 ratio, which suggests that many modern core top samples actually have contributions from deep-dwelling archaea (Van der Weijst et al., 2022). Thus, the

ambient temperature of Thaumarchaeota, which determine the cyclisation of GDGTs, is not strictly sea surface temperature. Even though the GDGTs may to variable extent derive from around the thermocline, it was shown that that SST has a strong relationship with surface temperature (Van der Weijst et al., 2022). Subsurface calibrations (Tierney and Tingley, 2014; Kim et al., 2015; Ho and Laepple, 2016) use variable ways to integrate temperature over depth, which still induce uncertainty on their reliability. Nonetheless, even though a perfect calibration does not exist yet, $TEX_{86}$ is still a valuable proxy that reflects

the temperature of a relatively stable layer of the ocean (e.g. Kim et al., 2016; Hurley et al., 2018) and provides a robust ocean temperature change in both trend and variability (Van der Weijst et al., 2022), especially when it is used along with other temperature proxies (e.g. Super et al., 2020, Leutert et al., 2020), which we here use $U^{k'}_{37}$ as well.

Here, we apply the spatial linear Bayesian calibration BAYSPAR (Tierney and Tingley, 2014; 2015) to translate $TEX_{86}$ values into temperatures using both surface (0–20 m) and depth-integrated temperature (0–200 m) calibrations (prior mean of 20°C,

prior standard deviation of 20°C). We applied the $U^{k'}_{37}$ paleothermometer based on alkenones as independent additional paleothermometer. $U^{k'}_{37}$ index values were calculated following Prahl and Wakeham (1987) and converted to SST using the BAYSPLINE calibration of Tierney and Tingley (2018) (prior standard deviation of 10°C). These two proxies together with the $TEX_{86}$ related overprint indices are combined to determine the sea (sub)surface temperature (SST) change.



## 3. Results

### 3.1 GDGT-based temperature reconstruction

The concentrations of all GDGTs are consistently high (~50 µg/g sediment for total isoGDGTs) in the early Miocene, and show a normal relative distribution, except for the interval around the MCO (287–256 mbsf). The isoGDGT concentration drops to 5 ug/g sediment at the onset of the MCO and remains stable until 7 Ma. In the MCO interval, of all GDGTs, crenarchaeol (Cren) and crenarchaeol (Cren') decreases the strongest (to 1/400, GDGT-3 decreases to 1/200, GDGT-2 and GDGT-1 decreases to 1/100, GDGT-0 decreases to 1/25; Suppl. Fig. 1). As Cren is not in the $TEX_{86}$ index, its anomalous trends in abundance do not directly affect $TEX_{86}$ values but do affect the GDGT indices and ratios that have Cren in the denominator. As a result of the extra decline in Cren, GDGT-0/Cren, MI and GDGT-2/Cren all yield abnormally high values in the MCO interval (Fig. 3). The GDGT-2/3 ratio is gradually increasing from 5 to 8 throughout the record, with transient peaks in the early Miocene and MCO. Cut-off values for this ratio vary among users and sites, between 3 and 10 (e.g., Bijl et al., 2021; Van der Weijst et al., 2022; Leutert et al., 2020). Hurley et al. (2018) demonstrated that the GDGT-2/GDGT-3 values rapidly rise from 3–5 in the surface mixed layer (upper 150m) to 20–25 at the thermocline depth (see also Basse et al., 2014; Hernández-Sánchez et al., 2014; Kim et al., 2016; Van der Weijst et al., 2022). In any case, sediments with GDGT-2/GDGT-3 values >3 might to some extent be biased towards deeper waters and, thus, to lower temperatures. ΔRing Index varied from -2.2 to 1 in the whole record and 152 data points fall outside the 95% confidence interval of the $RI-TEX_{86}$ array (Suppl. Fig. 3).

BIT index values show a large range of variation, between 0.1 and 0.9, and show a prominent peak (~ 0.9) during the MCO, and consistent low values (~0.1) in the Pliocene, indicating a potentially large contribution of GDGTs from land (Fig. 3). However, the #$rings_{tetra}$ values highly varied throughout time, but are consistently elevated between 17 and 7 Ma from 0.3 to more than 1.0, suggesting that brGDGTs have an in situ marine origin. In a ternary diagram of the tetra-, penta-, and hexa-methylated brGDGTs, Site 1168 samples plot generally offset to the global soil cluster (Suppl. Fig. 4), which also supports a non-soil origin. This would imply that the BIT index cannot be interpreted as indicator for the input of terrestrial matter at this site.

$TEX_{86}$ values of the early Miocene were around 0.65 but fluctuating, then reached 0.8 at 16 Ma in the MCO interval, although these values are probably affected by non-thermal overprints. There is an abrupt decline in $TEX_{86}$ values after the MCO and then a long-term decrease to 0.4 until 5 Ma. $TEX_{86}$ values increased to 0.6 in the early Pliocene, then decreased to 0.36 at 1.35 Ma, and eventually increased to 0.52 in the youngest sediment (Fig. 3). SSTs derived from the $TEX_{86}$ are around 25°C in the early Miocene section. $TEX_{86}$ values at the peak MCO would equate to SSTs of 34°C, followed by first a rapid cooling at





around 14.5 Ma, and then more gradual down to 7°C towards the end of the Miocene (Fig. 4). After an ephemeral warming in the early Pliocene to 20°C, SST decreased to 6°C in the mid-Pleistocene, and then recovered to the modern level around 17°C. Throughout the record, the difference between surface and depth-integrated sub-surface temperature derived from $TEX_{86}$ is

small (< 2°C).

### 3.2 Alkenone-based temperature reconstruction

The $U^{k'}_{37}$ index record varies between 0.43 to 0.93, and generally follows the trends of $TEX_{86}$, except during the MCO (Fig. 4). Early Miocene sediments have an average $U^{k'}_{37}$ value of ~0.8, in agreement with previously published data from the same site (Guitián and Stoll, 2021). In the MCO interval, $U^{k'}_{37}$ rose to 0.93. None of the analyzed sediments in the MCO interval

have saturated $U^{k'}_{37}$ index values. Subsequently, $U^{k'}_{37}$ gradually drops to 0.43 in the end of the Miocene and recovers to ~0.6 in the Pliocene. $U^{k'}_{37}$-based SST reveal similar temperatures to $TEX_{86}$-based SubT except in the MCO interval (Fig. 4). In the early Miocene, $SST_{UK37}$ yields ~24°C on average, then increases to ~28°C in the MCO. Subsequently $SST_{UK37}$ cools down to 10°C at 5 Ma and increases to 16°C in the Pliocene.



**Figure 3. TEX$_{86}$ values and indices/ratios to detect non-thermal GDGT contributions. Red dashed lines indicate proposed threshold values. a. TEX$_{86}$. b. Methane index, threshold=0.4 (Zhang et al., 2011) c. GDGT-2/Cren, threshold=0.4 (Weijers et al., 2011). d. GDGT-0/Cren, threshold=2 (Blaga et al., 2009). f. BIT, usually applied threshold=0.3 (Hopmans et al., 2004). e. GDGT-2/GDGT-3, threshold=5 (Taylor et al., 2013). g.#rings$_{tetra}$, threshold=0.8 (Sinninghe Damsté, 2016). Discarded data are shown by crosses.**


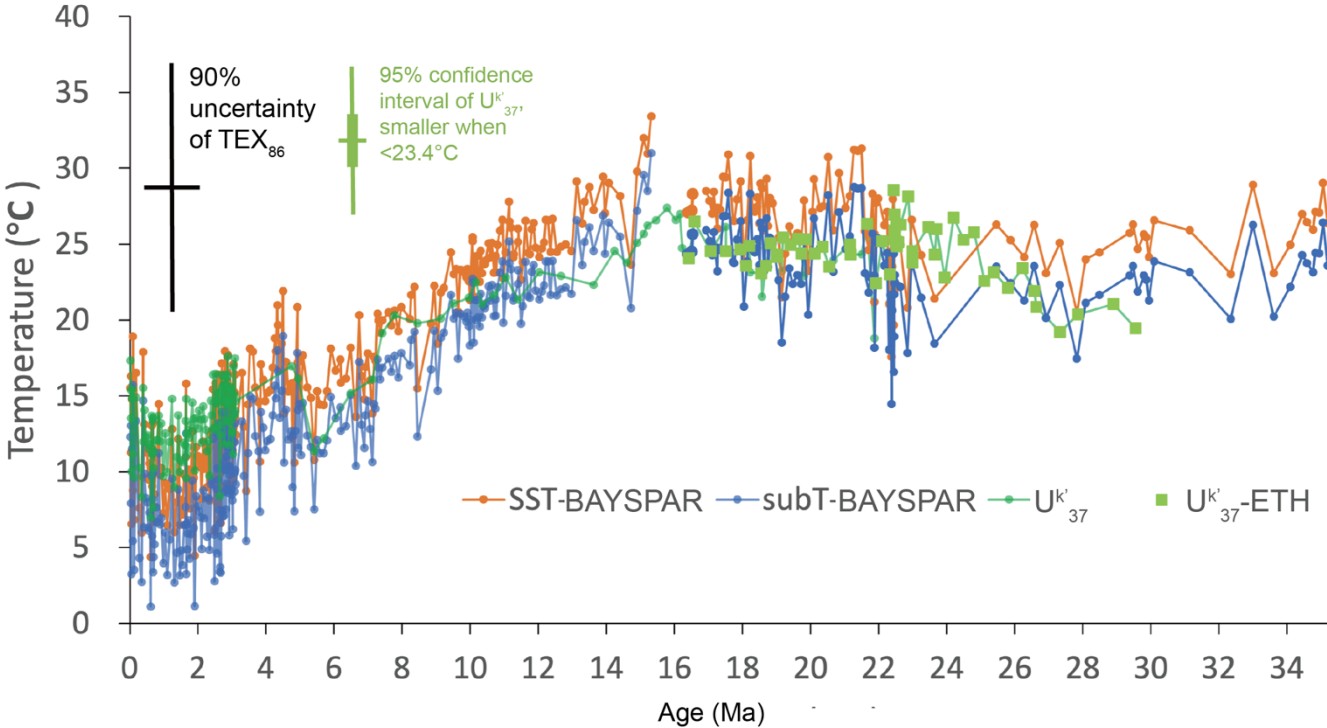

**Figure 4. TEX$_{86}$ and U$^{k'}_{37}$ SST reconstructions of Site 1168. The BAYSPAR calibration (Tierney and Tingley, 2014) is used to translate TEX$_{86}$ values into surface (orange points) and subsurface (blue points) temperatures. U$^{k'}_{37}$ SST reconstruction (dark green dots) based on the BAYSPLINE calibration (Tierney and Tingley, 2018). Oligocene–early Miocene U$^{k'}_{37}$ SST of Guitián and Stoll (2021; green squares).**


## 4. Discussion



## 4.1 Reliability assessment of the SST record

In all intervals of the records besides the MCO, temperature estimates derived from the $U^{k'}_{37}$ and $TEX_{86}$ have similar trends and absolute temperatures, demonstrating that both proxies represent the same water layer. The large $U^{k'}_{37}/TEX_{86}$ discrepancy

during the MCO suggests that one of the paleotemperature proxies is affected by non-thermal overprints (Fig. 4, 5). Given the many indices that signal anomalous isoGDGT distributions in the MCO interval, it is likely $TEX_{86}$ that has a non-pelagic GDGT assemblage and thus reflects an unreliable SST. However, the high GDGT-0/Cren, MI, GDGT-2/Cren and BIT index values in the MCO interval are all caused by the anomalously low contribution of Cren (i.e., the denominator) and not of elevated relative abundances of the signaling compound for that overprint (the nominator). Moreover, the low TOC wt%

(<0.5%) (Exon et al., 2001) in the MCO sediments does not support the existence of any cold seeps, anaerobic oxidation of methane or methane hydrate production, despite the high GDGT-0/Cren, MI and GDGT-2/Cren values (Fig. 3). If it is indeed the excess relative decrease in Cren concentration that causes the high BIT, GDGT-0/Cren, MI, GDGT-2/Cren and low RI, because all have Cren involved in their equation, the question is to what extent this affects $TEX_{86}$ values. In any case, the indices may not necessarily reflect the overprints that they are usually associated to at this site. However, since in the same

interval as the anomalous GDGT compositions $TEX_{86}$ and $U^{k'}_{37}$ disagree, and to be conservative, we discard the $TEX_{86}$-based SSTs with high confounding factor values in the MCO interval at this site.

The anomalously low relative abundance of Cren could be explained by different preservation efficiency and/or degradation rates for distinct GDGTs. The interval in which GDGT concentrations decrease (17.34–16.85 Ma, 294.66–274.28 mbsf) is concomitant to an interval of decreased terrigenous clay and quartz and increased calcium carbonate content (Fig. 5; Robert,

2004). The loss of terrigenous clay can lead to reduced preservation of organic matter (Ransom et al., 1998; Wu et al., 2019) because enhanced pore-water flow in the overlying sediment enhances oxygen exposure time (Huguet et al., 2008; Schouten et al., 2013). It was shown that isoGDGTs with more cyclopentane moieties are less resistant to oxidation than those with less cyclopentanes (Ding et al., 2013). As a result, degradation processes would result in lower $TEX_{86}$ values and an underestimation of SSTs. However, in the MCO interval $TEX_{86}$ was very high and led to SST reconstructions much higher

than $SST_{UK37}$. While selective degradation could be an explanation for the relatively excessive loss of Cren, it cannot explain the anomalously high $TEX_{86}$-based SSTs in that interval. On the other hand, preferential degradation of alkenones could have biased $U^{k'}_{37}$ at this site as well, notably in the form of a warm bias in SST (Freitas et al., 2017). Yet, in the MCO interval, we do not observe anomalous warmth, and the index is not yet saturated. Therefore, the selective degradation in alkenones cannot explain the SST difference between SSTs derived from $U^{k'}_{37}$ and $TEX_{86}$.





The extremely high (up to 100) GDGT-2/GDGT-3 ratio found in the MCO is unprecedented in both paleo and modern ocean records (e.g., Taylor et al., 2013; Hernández-Sánchez et al., 2014; Hurley et al., 2018; Besseling et al., 2019; Bijl et al., 2021; Van der Weijst et al., 2022). Perhaps the isoGDGTs found in the MCO interval were produced by other archaeal communities, e.g., Marine Group II and/or III (Besseling et al., 2019) and the ratio could have been further increased through selective degradation of GDGT-3 over GDGT-2. Given the temperature trend in the MCO interval, these GDGT producers may have

also responded to water temperature, although indirectly or with a different dependency. In the early and late Miocene, GDGT-2/GDGT-3 values are still relatively high (>5), indicating some isoGDGT input from deep water sources. The overall input of deeper-dwelling GDGTs may bias the reconstruction of absolute SST from $TEX_{86}$, but because the GDGT2/3 ratio in our post MCT interval is stable and without a trend, the trend of $TEX_{86}$ should not be influenced (Ho and Laepple, 2016; Leutert et al., 2020; Van der Weijst et al., 2022) and the amplitude is well constrained by $U^{k'}_{37}$. Thus, for the late Miocene and Pliocene the

sediments are not discarded, despite high GDGT-2/GDGT-3 values. Despite nearly half of the data points falling out of the 95% confidence interval of $TEX_{86}$-Ring Index, we decided to not discard those because most of the sediments with abnormal ring index values are caused by the highly reduced contribution of Cren.

Overall, considering the small difference (~2°C) between surface and subsurface calibration of $TEX_{86}$ and $SST_{UK37}$, the relatively large calibration error of proxies and similar extent of variability and the high GDGT-2/GDGT-3 ratio throughout

the study interval, we deem that both proxies mainly reflect temperature of the surface layer, with $TEX_{86}$ integrating a deeper component. Hence, we claim our temperature record a sea (sub)surface temperature (SST). But we focus on the $U^{k'}_{37}$ record when we interpret the record during the MCO.





**Figure 5: Lithology, GDGT concentrations and relative distributions and SST change in the mid-Miocene. a. Benthic foraminiferal**

**$\delta^{18}O$ compilation (Westerhold et al., 2020) b. SST changes based on $TEX_{86}$ and $U^{k'}_{37}$ (this study and Guitián and Stoll, 2020). Orange**

**dots indicate data points that are considered reliable and crosses indicate data points of $TEX_{86}$ that are considered unreliable c.**

**absolute abundance of Cren as indicator of GDGT preservation. d. Weight% calcium carbonate indicating lithology change,**



inversely related to clay content (Robert, 2004). Orange dashed line indicates the change of lithology, while isoGDGT preservation changes occur at the onset of MCO. Blue dash line indicates the SST warming, postdating the MCO. Yellow bar indicates the interval

of MCO.







**Figure 6: a. Paleolatitude reconstruction using GPlates (Torsvik et al., 2012; Van Hinsbergen et al., 2015) of Site 1168 (this study, Guitián and Stoll, 2021), Site 806 (Zhang et al., 2014), Site U1461 (He et al., 2021), Site 1171 (Leutert et al., 2020), Site U1459 (De Vleeschouwer et al., 2019), Site 594, Site 1125 (Herbert et al., 2016), Site U1356 (Hartman et al., 2018; Sangiorgi et al., 2018), the**

**Ross Sea compilation (AND-1B, AND-2A, DSDP 274, DSDP 270, CIROS 1, CRP 2/2A; McKay et al., 2012; Levy et al., 2016; Sangiorgi et al., 2020; Duncan et al., 2022) b. Reconstructed SST of the same sites using BAYSPAR and OPTiMAL (Ross Sea only) calibrations for TEX$_{86}$ (no symbol) and BAYSPLINE for U$^{k'}_{37}$ (triangles) and bottom water temperature based on benthic foraminiferal $\delta^{18}$O (Gaskell et al., 2022). c. Benthic foraminiferal $\delta^{18}$O compilation (Westerhold et al., 2020). Modern SSTs of the sites are indicated by the coloured stars at 0 Ma.**

**4.2 Site1168 SST evolution and Southern Ocean temperature gradient in the Neogene**

The new SST record for Site 1168 shows in broad lines a similar trend to the global compilation of benthic foraminiferal oxygen isotope stack ($\delta^{18}$O$_{bf}$), however there are some interesting deviations (Fig. 6). The mid-to-late Miocene interval contains a remarkable ~10°C gradual SST cooling that is much less prominent in $\delta^{18}$O$_{bf}$ (Fig. 6). Other than the late Miocene cooling, we have found that subtropical SSTs fluctuated around 26°C in the early Miocene, were slightly elevated in the MCO

and rapidly cooled 5°C across the MCT. Pliocene and Pleistocene SSTs at Site 1168 have larger variability than those in the late Miocene. But the variability remains in the Pliocene and Pleistocene, which deviates from $\delta^{18}$O$_{bf}$ feature.

The northward movement of the site, from ~52°S in the early Miocene to ~42°S at present, may have dampened the amplitude of Neogene long-term cooling to an unknown extent. However, during the Oligocene, ocean conditions at this site also barely changed, despite the northward drift, likely because the ocean currents migrated northwards along with the tectonic drift of

Australia (Hoem et al., 2021). For the same reason, northward tectonic drift of Australia during the Neogene, in other words the latitude change, may have had similarly little effect on the temperature evolution at this site, thus records the temperature resulting in both global climate and water mass change at Site 1168. The synchronous tectonic drift of other mid-latitude sites warrants the conclusion about latitudinal temperature gradient drawn from the comparison (Fig. 6a). With the consistency of both paleotemperature proxy results in consideration, we will further discuss the SST evolution per time interval, focusing on

variability within the record, comparison to the $\delta^{18}$O$_{bf}$ as representation of deep-sea temperature and global ice volume trends, and comparison to other SST records in the region to reconstruct latitudinal SST gradients.

**Early Miocene (23.04–17.0 Ma)**





In the early Miocene, SST was around 26°C, but punctuated by several short cooling events (Fig. 6). SST minima occurred at

22.4 Ma, 19.5 Ma and 17 Ma, roughly time-equivalent to ephemeral positive excursion events (Mi-1.1, Mi-1a, Mi-1b) in $\delta^{18}O_{bf}$ (Miller et al., 1991; Billups et al., 2002; Liebrand et al., 2011; Westerhold et al., 2020). SSTs from the Wilkes Land margin (U1356) reflect similar events at 22.4 Ma and 17 Ma (Sangiorgi et al., 2018; Hartman et al., 2018). In contrast, SSTs in the Ross Sea remained relatively stable and profoundly cooler than U1356, around 4°C using the OPTiMAL calibration (Duncan et al., 2022). We choose OPTiMAL as the calibration for the Ross Sea sites because the Ross Sea has experienced glacial

phases in the early and middle Miocene (Passchier et al., 2011; Marschalek et al., 2021) while the Wilkes Land was continued to be surrounded by warm oligotrophic waters (Bijl et al., 2018; Sangiorgi et al., 2018).

The latitudinal SST gradient between the STF and the higher latitudes was relatively constant during the early Miocene, remaining around 9°C (Fig. 7). This gradient is very similar to the modern gradient between 51°S and Antarctic margin (10°C; Fig. 1, 7; Hartman et al., 2018), and represents a similar gradient to that of the late Oligocene (Hoem et al., 2022). Such a

gradient may testify to the presence of a relatively strong proto-ACC when the Tasmanian Gateway aligned to the westerly winds (Scher et al., 2015; Pfuhl et al., 2004; Sauermilch et al., 2021), although the absolute SSTs at both sites were higher in the Miocene than today (Fig. 6).

A recent study (Kim and Zhang, 2022) suggested that a massive methane hydrate destabilization event took place at the south Australian Margin during the Oligocene-Miocene boundary based on an elevated MI and more negative compound specific

carbon isotopes of Site 1168. However, based on the age model of Stickley et al. (2004) calibrated to GTS 2020, the Oligocene-Miocene boundary indicated by Kim and Zhang (2022) at ~416 mbsf is actually around 22.6 Ma. On the other hand, the high MI is actually induced by less Cren and more GDGT-0, rather than an increase in GDGT-1,2,3 (Suppl. Fig. 1) which are thought to be produced by methanotrophic archaea. Thus, we doubt if the evidence is concrete enough to prove a major methane hydrates dissociation in the early Miocene but acknowledge their hypothesis.


**MCO (17.0–14.5 Ma)**

$SST_{UK37}$ shows a slight warming of ~2 degrees to ~27°C at the onset of the MCO (Fig.5, 6). As this warming cannot be ascribed to saturation of the proxy, this would mean that the SST increase during the MCO is indeed smaller at the STF than at high latitude sites, and than what would be assumed from the strong change in $\delta^{18}O_{bf}$ at this time. Still, also at Site 1168, the mid-

Miocene stands out as a warm time interval, consistent with other records, both surface and bottom (Levy et al., 2016; Sangiorgi et al., 2018; Modestou et al., 2020). Compared to the clear trends in the global $\delta^{18}O_{bf}$ record (Westerhold et al., 2020), the onset of the MCO is less clearly expressed in the records of SST (this study, Shevenell et al., 2004; Levy et al., 2016; Hartman



et al., 2018; Super et al., 2018; 2020) and Mg/Ca based bottom water temperature (Lear et al., 2015). Hence, the relationships between changes in surface oceanography, ice volume, and deep-sea temperature at the MCO onset remain unresolved at this

stage. In addition, the currently available clumped isotope records do not fully capture the 16–17 Ma interval (Modestou et al., 2020; Meckler et al., 2022) which makes disentangling ice volume and deep-sea temperature effects in $\delta^{18}O_{bf}$ at the MCO onset problematic.

The lithological change of Site 1168 at the onset of the MCO coincides with the biomarker preservation change, but precedes regional warming and $\delta^{18}O_{bf}$ decline (Fig. 5). The increase of calcium carbonate indicates a change in the depositional setting,

likely due to an intensification of the Proto-Leeuwin Current (PLC). The intensification of the PLC during MCO is interpreted from deepest seabed scouring (Jackson et al., 2019) as well as the existence of tropical corals and 'larger foraminifera' (McGowran et al., 1997) in the Great Australian Bight (Fig. 1). Besides the increase of surface calcite productivity, this led to better ventilated bottom water and warm oligotrophic surface water conditions during MCO. The potentially limited preservation of GDGTs may be related to improved bottom water oxygenation (Huguet et al., 2008).

The SST records from Site 1168, near the subtropical front, and Site 1171, in the subantarctic zone, suggest that the latitudinal SST gradient collapsed to 0 (Fig. 7). This implies that the latitudinal SST gradient across the Tasmanian Gateway was strongly reduced in the MCO. The reduced latitudinal temperature gradient persisted both equatorward and poleward. Similarly, in the eastern Equatorial Pacific (Rousselle et al., 2013), the latitudinal SST difference between the equator and subtropical region was reduced to ~2°C. We also note that the SSTs of Site 1168 and high latitude sites (Wilkes land, Ross Sea) are close to each

other (Fig. 7), even though SST reconstructions of high latitude sites are sparse (U1356) and absolute values are highly dependent on the used calibration (Ross Sea sites) (Fig. 6). The reduced latitudinal SST gradient between mid-latitude and polar region (Fig. 6, 7) agrees with certain modeled results for the MCO (Herold et al., 2011; 2012) but is not captured by other modeling (Burls et al., 2021). The weakened latitudinal SST gradient also means that the proto-ACC was reduced in strength (Sangiorgi et al., 2018) compared to during the early Miocene.





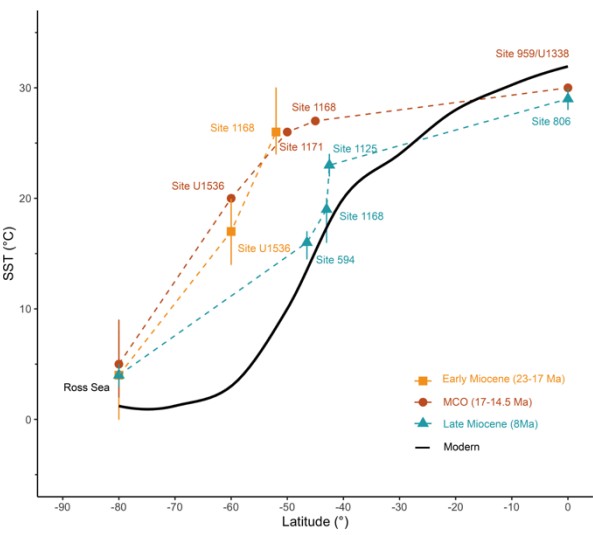

**Figure 7: Sea (sub)surface latitudinal temperature gradient in the southern hemisphere, of the early Miocene (orange; 23-17 Ma), MCO (red; 17-14.5 Ma), late Miocene (blue; 8 Ma) and modern at 160°E (black; based on Fig. 1d), respectively. Error bars indicate the variability of the time range. Paleo data are from the sources in Figure 6 except for the MCO SST at equator, which are from Rouselle et al. (2013), Van der Weijst et al. (2021).**

**MCT (14.5–12.5 Ma)**

The termination of the MCO is reflected by a sharp 5°C decrease in SST at Site 1168 from 14.5 to 13 Ma, coincident with the time of cooling at nearby Site 1171 (Leutert et al., 2020) (Fig. 6). This cooling phase coincided with a strong increase in $\delta^{18}O_{bf}$, which mostly reflects Antarctic ice sheet expansion (Shevenell et al., 2004, 2008; Leutert et al., 2020) and potential northward expansion of subantarctic waters (Leutert et al., 2020) accompanied by $pCO_2$ decline (Super et al., 2018). The SST gradient between the mid- (Site 1168 and Site 1171) and high latitudes (Site U1356) increased, suggesting stronger fronts with amplified cooling towards Antarctica (Fig. 6). This is most likely a result of polar amplification of cooling towards high latitudes, further exacerbated by the expanding ice sheet and northward migration of frontal systems.

**Late Miocene (12.5–5.3 Ma)**

After the MCT, SSTs at Site 1168 gradually cooled by 10 °C from 13 to 5 Ma and the short-term variability amplitude was notably small (2–3°C) (Fig. 6). The amplitude of this cooling is comparable to that at other mid-latitude sites, i.e., Site 594 and Site 1125 in the southwest Pacific (Herbert et al., 2016).



The absolute temperatures and the cooling trend in the SST record of Site 594 seem in agreement with those at Site 1171,

which is located at the same latitude and bathed by the same subantarctic water in the modern system. Thus, we can consider

Site 1171 and Site 594 as one continuous record representing the ocean temperature 5° south to Site 1168. Furthermore, the

temperature difference between closely located subtropical (Site 1125, Site 1168) and subantarctic (Site 594, Site 1171) sites

became larger, about 4°C between each area (Fig. 6, 7). We deduce from this that the STF progressively got stronger. The

equator-mid latitude temperature gradient also progressively increased (Fig. 7). Equatorial SST of Site 806 in the west Pacific

decreased only 3°C from 12 Ma to 5 Ma (Zhang et al., 2014). Given the low temperatures at the high latitudes by the end of

the Miocene (Gaskell et al., 2022) (Fig. 6), the Southern Ocean equator-to-pole latitudinal temperature gradient and the SSTs

of the studied sites must have been very similar (~26°C) to modern conditions. Notably, the relatively strong Southern Ocean

cooling trend is not reflected in the $\delta^{18}O_{bf}$ record, which remained relatively stable over this time interval. Thus, the relationship

between Southern Ocean cooling, deep sea temperature change, and ice volume change still needs further study for this interval.

The increased SST gradient reflects a combination of global cooling and the amplification effect of northward migration of the

STF, which effectively stretches the Southern Ocean over wider latitudes. This change in temperature gradient between

equatorial sites and Site 1168 also indicates a weakening of the Leeuwin current (De Vleeschouwer et al., 2019). The gradually

increased equator to mid-high latitude SST gradient led to a contracted and strengthened Hadley cell, which consequently

caused an aridification in subtropical region by intensifying the evaporation in the descending limb (Herbert et al., 2016;

Groeneveld et al., 2017). This is reflected at Site 1168 and other Tasmanian sites with an increase in the contribution of kaolinite

and/or illite, reflecting erosion of old soils as a result of aridification of the hinterland and subsequent transport by the westerlies

since the late Miocene (since ~200 mbsf of Site 1168; Robert, 2004).

During 8–5.3 Ma, several stepwise changes occur in the Southern Ocean. Tanner et al. (2020) indicated a northward STF shift

at the Agulhas ridge from 8 Ma onwards, while Groeneveld et al. (2017) suggested a southward migration of the westerlies

near southwest Australia in the same period. This apparent discrepancy can be partly accounted for with the northward tectonic

movement of Australia while the STF remained at its position, or moved northward at a slower pace than the continent (De

Vleeschouwer et al., 2019), while the Agulhas plateau experienced limited tectonic movement. This would mean that Site 1168

entered the subtropical zone, received more warm subtropical water through Leeuwin current, and therefore reduced the

temperature difference between west Australian coastal sites. However, the increased SST difference that we observe between

Site 1168 and northwest Australia/equator (Fig. 6) suggest a weaker PLC, and thus does not seem to support this interpretation.

Another explanation of the discrepancy between Tanner et al. (2020) and Groeneveld et al. (2017) is that the STF did not





necessarily align to the position of the westerlies (De Boer et al., 2013). The relative positional shift between the Australian continent and the westerlies does not directly influence the position of the STF when the STF remains bound by the southern edge of the Australian continent during this time.


**Pliocene to modern (5.3–0 Ma)**

Following late Miocene cooling from its minimum in the latest Miocene (Herbert et al., 2016), the Pliocene SSTs of Site 1168 shifted back to generally warm conditions (16 °C) based on the $TEX_{86}$ (Fig. 6). Compared to the modest variability in $\delta^{18}O_{bf}$ in the Pliocene, SST change at the STF is relatively large in amplitude (8°C). Pliocene SST variability at Site 1168 may be

amplified by the combined effects of a changing Leeuwin current, migrating STF, and strong ice sheet fluctuation on Antarctica, and the variability would then be stronger than during the Miocene. SST shifts can be roughly correlated to known $\delta^{18}O_{bf}$ events, such as mid Piacenzian Warm Period and M2 glaciation. Intriguingly, in the Pliocene-Pleistocene interval, the SST of Site 1168 varied synchronously with that in the Ross Sea, which lead to a constant temperature gradient of ~8°C between mid- and high latitudes.


In summary, the Neogene SST record of the STF is characterized by 5 phases of accelerated cooling of ~2–3 degrees cooling at ~14 Ma, 9 Ma, 7 Ma, 5.5 Ma and 2.8 Ma, superimposed on the gradual cooling trend (Fig. 6). Overall, by comparing our mid-latitude SST record with other sites in the southern hemisphere, we show that the latitudinal gradient of the Southern Ocean varied from weak (23–17Ma), to weakened (17–14.5 Ma), to strengthening (14.5–5 Ma), to variable (5–0 Ma) (Fig. 7)

which we link to the gradual development of the frontal systems in the Southern Ocean, related to the interplay between ice sheet, tectonic and climatic evolution of the Neogene Southern Ocean. The long-term evolution of subtropical front SSTs is consistent with that of $\delta^{18}O_{bf}$, except for the progressive 10°C cooling in the late Miocene that is less pronounced in $\delta^{18}O_{bf}$ (~0.3 permille) (Fig. 6). Perhaps this reflects a nonlinear response of subtropical front SST to the progressive buildup of ice sheets once it has passed a critical threshold, including that of the west-Antarctic ice sheet (Marschalek et al., 2021), and a

more persistent presence of sea ice (Sangiorgi et al., 2018). In general, we observe modest cooling in the equatorial Pacific region throughout the Neogene, and modest cooling also at the Antarctic continental margin. But the strong cooling in the subtropical front region suggests that the shape of the meridional temperature gradient changed fundamentally, with broad warm Southern Ocean in the mid-Miocene and progressive expansion of cold-temperate conditions towards lower latitudes thereafter. This could be caused by a progressive increase in strength of the ACC, strengthening latitudinal SST gradients in

mid-to-low latitudes. Further study should reveal the exact changes of both surface oceanographic conditions (not exclusively SST but also upwelling and salinity) and deep-water temperature (including separate deep-water temperature and ice volume signals).

## 5. Conclusion

Our Neogene SST record from offshore Tasmania, derived from 2 independent biomarker proxies, provides a first continuous, long-term record of subtropical Southern Ocean SST evolution during the gateway opening. The SST record reflects a warm mid-Miocene climatic optimum, a gradual but profound 10°C cooling in the mid-to-late Miocene and renewed warming in the Pliocene and highly variable temperature conditions during the Pleistocene to modern times. The long-term SST trend is consistent with the $\delta^{18}O_{bf}$ record, except during the late Miocene cooling. Short-term SST variability in the record can be linked 410 to glacial-interglacial phases, suggesting strong coupling between Antarctic ice sheet buildup and the subtropical front temperature on a multi-million-year timescale. However, the mechanism of the decoupling between SST and $\delta^{18}O_{bf}$ during the late Miocene is still enigmatic. Comparison with previously published SST records indicates that the latitudinal gradient in the Southern Ocean experienced a stepwise development, from a relatively strong gradient in the early Miocene, to reduced gradients during the MCO, and then steepening during the mid-to-late Miocene. Pliocene to modern time gradients remain 415 relatively constant in trend but may differ in range over glacial-interglacial cycles. During the late Miocene cooling, the latitudinal SST gradient between the STF and Pacific equator profoundly increased (from 4°C to 14°C) while the SST gradient between the STF and the Antarctic margin decreased due to amplified STF cooling and relatively stable near-Antarctic SSTs. This caused a progressive narrowing of the warm equatorial-to-subtropical heat distribution, and an expansion of subpolar conditions to lower latitudes. Our study presents a continuous picture of the STF temperature evolution in the Southern Ocean 420 Tasmanian gateway area and reveals the history of frontal systems towards modern conditions.

**Data availability:**

TEX$_{86}$ and U$^{k'}$37 data of ODP Site 1168 is deposited at Zenodo https://doi.org/10.5281/zenodo.7119904.



**Competing interests:**

The authors declare that they have no conflict of interest

**Author contributions:**

PKB designed the research. SH, FH and FL processed samples for organic geochemistry (TEX$_{86}$ and U$^{k'}_{37}$). All authors contributed to analysing the data. SH designed the figures and wrote the paper with input from all authors.

**Acknowledgements:**

We thank Mariska Hoorweg, Klaas Nierop and Desmond Eefting for laboratory assistance. We thank Johan Weijers, Benjamin Petrick, Guodong Jia, Stefan Schouten, Robert McKay and Bella Duncan for providing corresponding data which were used to interpret the results. We thank IODP and shipboard scientists of ODP 189, especially KCC in Japan for the help with sampling. Additional gratitude is sent to Kun Huang for helping generate Monte-Carlo simulation of GDGTs compositions although not included in the manuscript and Mei Nelissen for preliminary data analysis. This research is funded by ERC Starting Grant 802835 to Peter K. Bijl.

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
