# Peer review of "Lipid biomarker-based sea surface temperature record offshore"

_Climate of the Past, 2022_

## Author Response (AR1)

**Reply to RC1**

**Comment on cp-2022-79**

Igor Obrecht (Referee)

Referee comment on "Lipid biomarker-based sea (sub)surface temperature record offshore Tasmania over the last 23 million years" by Suning Hou et al., Clim. Past Discuss., https://doi.org/10.5194/cp-2022-79-RC1, 2022

I read with great interest the study submitted by Hou et al. "Lipid biomarker-based sea (sub)surface temperature record offshore Tasmania over the last 23 million years". What a nice study - well designed, well written, clear and to the point. Well done.

Response to the reviewer: We thank reviewer 1 (Igor Obrecht) for his interest in our work and the effort he put into the referee report.

However, I would have a few remarks.

1 Minor comment: It is a bit confusing that you refer even in the title with (sub)surface temperatures, and then follow in the abstract and introduction, but explain why your data could be representative of subsurface temperature only around line 125. Please clarify why you assume your data is representative of subsurface temperature (if I understand correctly only one proxy is assumed to be potentially affected by this). Alternatively, you might just say "sea surface temperature proxy record", and later explain that one of your proxies is potentially biased to subsurface temperatures. Also, by having SST as an abbreviation for sea (sub)surface temperature, it is unclear when you refer to alkenone-based SST if you also consider it as potentially subsurface temperature.

Response to the reviewer: Thank you for pointing this out. We initially used the term (sub)surface because we cannot rule out a subsurface contribution of the GDGTs even though TEX86-based temperature is strongly related to the sea surface temperature (see explanation in lines 125-130). Upon revising the manuscript we now agree that the term (sub)surface may generate more confusion than necessary. We will now use sea surface temperature instead, and keep the nuance for a more elaborate consideration in the text.

Proposed changes in the manuscript: We will more clearly explain the considerations on what kind of temperature TEX86 reflects into the manuscript (specifically under "methods"), and replace "sea (sub)surface temperature" by "sea surface temperature" throughout.

2 Major comment: I am a bit disappointed that the authors didn't discuss Pleistocene data, but rather squeezed it into the section Pliocene to modern. Even more surprising is that they made some comments about this part in the concussions without mentioning Pleistocene in Discussion (?!). Ideally, I would like to see some more thoughts on this interval. This is the part of the record with (not surprisingly) most of the variability and I would like in particular to see more discussion about:

Response to the reviewer: We understand the point raised by the reviewer and agree that the

Pleistocene is interesting. However, we believe that a detailed focus on the Pleistocene glacial-interglacial cycles is beyond the scope of this study, which rather aims to reconstruct the long-term evolution of the SST of the subtropical front and meridional SST gradient in the Southern Ocean in relation to the buildup of ice sheets over the entire Neogene. Such long-term records are still relatively scarce, as also noted by reviewer #2. An in-depth evaluation of Pleistocene SST change around Tasmania, is planned as follow-up study.

Proposed changes in the manuscript: No changes made.

a) I would be happy to see a comparison of the variability between alkenone and GDGT-based SSTs in the Pleistocene record and see if they are in phase and if the amplitude is similar.

Response to the reviewer: This is a good suggestion of the reviewer

Proposed changes in the manuscript: We will compare in text the variability, amplitude of $TEX_{86}$ and $U^{k'}_{37}$ of the Pleistocene in section 3.2.

b) During this period, something interesting happens. While TEX86 SST is in a consistently in similar range or above Uk'37-based SST throughout most of the record, with the onset of the Pleistocene TEX86 drops significantly when compared to alkenone SST. Could authors give some of their thoughts on that?

Response to the reviewer: We have also noticed this interesting trend: until the end of the Miocene $TEX_{86}$ seems to indicate warmer SST than UK37, during the Pliocene and the Pleistocene the two proxies seem to better overlap. We were careful in interpreting this signal as calibration errors do continuously overlap. Moreover, the comparison between SSTs from both proxies is so much better than many other instances where both proxies were applied on the same samples, that even small deviations stand out in our record. Also the choice of calibration impact the correlation between the proxies. SST results using the BAYSPAR calibration is barely different from $TEX_{86}{}^{H}$ based on the temperature output when $TEX_{86} > 0.5$. However, when $TEX_{86}$ is smaller than 0.5, namely in the Pleistocene, $TEX_{86}{}^{H}$ -based SST is well in line with $SST_{UK37}$.

Proposed changes in the manuscript: We will present this discussion in section 4.1 in the revised manuscript.

3 Following on the last point, some discussion if TEX86 records subsurface SST in some parts of the record and SST in others. For example, is it possible that in the PleistoceneTEX86 records subsurface temperature, and in the older part surface?

Response to the reviewer: In a comment further up we propose to elaborate in the methods a bit more on the consideration of calibration and whether TEX86 is a sea surface and subsurface temperature proxy. Moreover, the extent to which subsurface GDGTs impact the signal can be

evaluated using the GDGT2/3 ratio (as explained in Section 4.1), which shows no correlation to TEX86 vs Uk'37 difference.

Proposed changes in the manuscript: no extra changes made

4 Minor comment: References in the text seem to be without consistent formatting. Either use alphabetical or chronological order. It is a bit messy right now.

Proposed changes in the manuscript: We will list the in-text references in the order of published year, from the oldest to the most recent. We will check the order throughout the manuscript.

5 Minor comment: Can you explain why did you decide to use these specific calibrations and why are they more suitable than others?

Response to the reviewer: We made a careful consideration of calibrations based on the following criteria: BAYSPAR calibration is barely different from $TEX_{86}^H$ based on the temperature output when $TEX_{86} > 0.5$. Similarly, the Uk'37 linear calibration (Prahl and Wakeham, 1987) does not yield different temperatures than BAYSPLINE for our dataset. We choose to use the Bayesian calibrations here because the spatial characteristics and high temperature applicability were considered.

Proposed changes in the manuscript: We now explain our reasoning for which calibration we chose more explicitly in the methods section. We will add the figure below showing the temperature records using different calibrations in the supplementary material as Fig. S6

6 Line 169-170: Please rephrase. From this, it sounds that based on TEX86 you can get a difference between surface and subsurface layers (and that you assume it is <2degree). What you refer here (I presume) is that based on the different calibrations, you get that TEX86 temperature reconstruction for surface layers is <2 degrees higher than when you calibrate the same TEX86 data to subsurface

calibration. Right?

Response to the reviewer:    Yes. We will rephrase.
Proposed changes in the manuscript: rephrased to: 'Throughout the record, the difference in temperature derived from surface and depth-integrated sub-surface calibrations respectively is small (< 2°C).'

7 Line 193-194. I am really not convinced by this statement (in particular I mean "Uk′37 and TEX86 have similar trends and absolute temperatures, demonstrating that both proxies represent the same water layer"). Why this demonstrates that they are from the same layer? The absolute SST uncertainty is so large that you cannot say if the absolute SST are in the same range or not. However, I agree that the similarity in the trends is remarkable (except MCO and Pleistocene), but this does not have to mean that the evolution of the sea surface and the subsurface temperature was not coherent throughout the record.

Proposed changes in the manuscript: rephrased to:'temperature estimates derived from the Uk′37 and TEX86 have similar trends and absolute temperatures, demonstrating that both proxies represent the upper layer of the water column (0-200m).'

8 Minor comment: I understand that BAYSPLINE calibration is better suited for high SST and probably this is why the authors used it here. However, can you comment on how reliable it can be when SST reach 28 degrees as during MCO? Could it be that it just simply cannot account for higher temperatures?

Response to the reviewer:    The Uk'37 proxy is not yet saturated during the MCO in our record (0.94, 27.4°C). therefore, we assume that it reflects the actual SST, and that SST was not much higher. Rather we assume that TEX86 is problematic in this interval because of the high Methane index, GDGD-2/3 ratio, GDGT-0/Cren etc., which are due to the selective oxidation of GDGTs.

Proposed changes in the manuscript: No changes were made.

9 Please define MCT in the text and MMCT in Fig. 6.

Proposed changes in the manuscript: We will define the mid-Miocene climatic transition as MMCT instead of MCT (14.5-12.5 Ma) throughout the manuscript and figures.

I (obviously) like the manuscript and from my side, I recommend this manuscript for acceptance after moderate to minor revisions.

Response to the reviewer: Thanks again. Your comments are really appreciated.

**Reply to RC2**

**Comment on cp-2022-79**

RE: Lipid biomarker-based sea (sub)surface temperature record offshore Tasmania over the last 23 million years

I read the article "Lipid biomarker-based sea (sub)surface temperature record offshore Tasmania over the last 23 million years". Overall I was very impressed with the record and the analysis of the different proxies used in this study. There are very few multi-million year multiproxy records, and the analysis of the different proxies is interesting and useful. Overall I think that with minor changes, this article is ready for publication.

Response to the reviewer: We thank RC2 Benjamin Petrick for the positive assessment of our manuscript

The main issue that needs correcting is that the authors focus on the proxy comparison but are missing the opportunity to explore some of the implications of the record in terms of its relationship to changes in global circulation, especially in the more recent parts of the record. However, this is a minor complaint, and as I will explore in more detail below, I think it can easily be fixed with a few minor changes.

Response to the reviewer: we will reply to this main point in detail below.

My more detailed corrections are as follows:

Title: I found the title confusing with the (sub)surface in there. Given that you do an excellent job explaining the differences, I think this is unnecessary to use an awkward term like this. I would lose the word and just call it a temperature record.

Proposed changes in the manuscript: As explained at the similar comment made by reviewer 1, we will use 'sea surface temperature' instead, and discuss any nuances of the temperature record in the method (calibration selection) and the results (implications of different calibrations and data interpretation).

Line 60-65: some kind of description of the average modern SST of the site and the seasonal range (i.e., winter/summer SSTs) would be helpful here. I know that SST is shown in figure 1 but given that SST reconstructions and differences between the various proxies are a crucial aspect of this paper, I would like to see how it compares to the modern day and if there are any seasonal changes.

Proposed changes in the manuscript: We will add information on the modern-day temperature to section 2.1 of the manuscript and rephrase as 'cold and fresh subantarctic water masses (Heath et al., 1985; Exon et al., 2001) with modern SSTs ranging from 13°C to 17°C (winter-summer)'.

Figure 1: The currents here change colors and sizes in the different figures, and I cannot tell if that is intentional. If it is, then the changes need to be more prominent. Also, the outline of the modern-day location of Australia might help see the differences in the plate movement during these different times.

Response to the reviewer: The currents do not change in colors, so this may be an unintended optical illusion. Only the thickness of the lines changes as a reflection the strength of the current, which is based on Evagelinos et al. (2021). The modern-day location of Australia is shown in Figure 1d.

Proposed changes in the manuscript: We will make the difference in the current lines clearer and explain in the caption explicitly what the thickness of the current lines reflects.

Line 171-176: I would like to see the 37/38 and 37et/38et ratios for the UK37' here. While I do not expect them to show anything, I have noticed issues in deeper water before, which may indicate changes in the Aleknone producers. Also, given all the work done on testing and demonstrating when the TEX86 reflects thermal changes, a similar, at least brief, analysis of the Alkenones is appropriate. These ratios could be summed up here, and the figures could go in the supplement.

Response to the reviewer: This is indeed a good point. Upon this comment, we have revisited the spectra and evaluated the presence and abundance of these components.

Proposed changes in the manuscript: We will add to the methods section 2.5 of the text:' In order to assess the influence of potential changes in the algae composition on the SST reconstruction, we explored the ratio between C37 and C38 (C37/C38) (Rosell-Melé. et al., 1994) and the ratio between all C37 and the ethyl C38 alkenones (C37/C38Et) (Zheng et al., 2019).'. We will add to the results section of the text:'C38 alkenones are mostly at or below detection limit in the sediments older than 8 Ma. In the younger sediments we find 4 clear peaks representing C38 alkenones. Here, C37/C38 fluctuates between 0.9 and 1.2 while C37/C38Et varies between 1.4 and 1.2 (Suppl. Fig. 5).' in section 3.2. We will add to the discussion of the text:'C37/C38 and C37/C38Et ratios do not show any profound change, thus we infer that alkenone composition is not affected by algae composition changes and thus UK'37 represents sea surface temperature' in section 4.1

Fig 4: the UK37' for the previous study is written as UK37'-ETH. Is this a different method of calculating SSTs, or is something else going on here? Please explain. Also, I got confused in the paper in general by what is new data and what was done before. A statement describing this a little clearer would be great.

Response to the reviewer: As is written in the figure caption, UK'37-ETH indicates the data from Guitián and Stoll (2021), which was measured at ETH in Zurich. But we understand that while the other colors/items in the panel legend are different calibrations, it looks like UK37-ETH represent a different calibration as well.

Proposed changes in the manuscript: We will adjust the panel legend. All UK'37 will have the same green fill and line color. Symbols still differentiate between ETH-lab-produced data and new data in this study. We will adjust the figure panel legend so that it is clear that all UK'37 data is on the

same calibration. We will add in the text 'Our new GDGT data is derived from 412.7 to 0 mbsf (22.6-0 Ma)'and 'Our new alkenone data is derived from 363 to 0 mbsf (21.9-0 Ma). Existing Uk'37 data from this site was published in (Guitián and Stoll, 2021)' into section 3.1 and 3.2 respectively.

Figure 6: sorry, I found this figure confusing (it may just be me). The top panel could easily be summed up or put into the supplement. There is not much plate movement. Also, is there some way to highlight the main record better? Right now, it is getting lost in the rest of the data.

Response to the reviewer: Although the panel with a 0-90° y-axis does not seem to show much change in paleolatitude, the movement is actually large (up to 10 degrees). As the latitudinal SST gradient changes are a key aspect in our paper, we chose to retain the paleolatitude panel. We have revisited the figure to improve the level of detail and clarity, and finally decided on bringing our new dataset (in black) to the front to make it stand out more in relation to the other records.

Proposed changes in the manuscript: move the new dataset (in black) to the front in the figure.

Line 255: a short line here about why you are not talking about data before 23 Ma. (Even if it was talked about previously, a refresher might be nice).

Response to the reviewer: The reason we did not include any discussion on the record >23 Ma is because this part is extensively discussed in Hoem et al. (2022, CE&E). In our manuscript, we added the older than 23Ma data to the results part of our manuscript only to compare overprinting indices between the two parts of the record.

Proposed changes in the manuscript: We will rephrase as 'Other than the late Miocene cooling, we have found that subtropical SSTs fluctuated around 26°C in the early Miocene, which is similar to those in the Oligocene (Fig. 4) (Guitián and Stoll, 2021; Hoem et al., 2022). The SST record from >23 Ma is elaborately discussed in Hoem et al. (2022) so we will focus on the SST trends in the interval <23 Ma'

Line 311: the presence of tropical corals in the Great Australian Bight is debated. It might be nice to add a word or two to show this.

Proposed changes in the manuscript: We toned this down to 'The reported existence of 'larger foraminifera' (Gouley and Gallagher, 2004) and, perhaps, although debated, tropical corals (McGowran et al., 1997) in the Great Australian Bight (Fig. 1) are likely related to the warm water induced by PLC.'

Line 324: A line here about the greater impactions of the reduced gradient might be nice. This seems to be something that would have a global impact.

Proposed changes in the manuscript: We will rephrase as 'The weakened latitudinal SST gradient means that the ACC and associated fronts were weaker and/or located closer to Antarctica. Meanwhile, enhanced evaporation led to more precipitation in the high latitudes, at least on the Antarctic continental margin (Sangiorgi et al., 2018). The global heat transport also likely weakened

with latitudinally more equable climates (Chiang, 2009).'

Line 380: this idea about the Leeuwin current driving higher variability seems to be placed here without much evidence to back it up. Given that there are Pliocene and Pleistocene records of Leeuwin current SST, this seems to be at least something that could be compared.

Response to the reviewer: We thank the reviewer for this useful suggestion.

Proposed changes in the manuscript: We will add 'The early Pliocene warmth found at site 1168 coincides with the SST record of the northwest Australian continental shelf (He et al., 2021), thus confirming the existence of a relatively strong Leeuwin current which causes the SST rise. The subsequent cooling after 4 Ma until the mid-Pleistocene led to an increased SST gradient between Site 1168 and west Australian coast sites along the Leeuwin current pathway, therefore, indicating a stepwise weakening of the Leeuwin current (De Vleeschouwer et al., 2019) (Fig. 6).'

I would like to see some discussion of the recent theory put forward by Christensen et al. (2021) on the impact of changes in Tasman Leakage on global circulation around 6-7 Ma. Given the location of this core, this is something you should address.

Response to the reviewer: The possible influence of changes in Tasman Leakage was considered during the drafting of this paper. However, linking this with our record is not straightforward, as the Tasman Leakage relates to intermediate waters and our paper focuses on seawater temperatures that are reconstructed based on algae and microbes that live in the upper 0-200 m of the water column, which represents a different water mass. Moreover, we do not see any notable change happening in the surface waters at the time of the onset of the Tasmanian leakage around 7 Ma. But we can mention this briefly.

Proposed changes in the manuscript: We will add 'Christensen et al. (2021) suggest that changes in Tasman Leakage around 7 Ma may have an influence on the global circulation, in accordance with a southward migration of the westerlies. However, the Tasman leakage represents an intermediate water layer (400-900 m), which would not necessarily have influenced SST. Indeed, the SST record does not show a prominent, step-wise change around the time of the onset of Tasman Leakage' In the end of section 4.2 Late Miocene.

I also feel like the other reviewer that the Pleistocene record is described too briefly. The record is higher resolution there, so it feels like some discussion of the impact of intensification of northern hemisphere glaciation or the MPT (see (Auer et al., 2021)) or even the different patterns in the TEX86 and UK37' during the 100 Ka cycles may be interesting. I feel this last part is not treated the same way as the earlier cycles.

Response to the reviewer: See answer to reviewer #1.

.

In conclusion, a bit more work needs to be put into the interpretation of the records. Otherwise, excellent job, and I look forward to seeing the final article. I recommend Minor corrections.

Cites

Auer, G., Petrick, B., Yoshimura, T., Mamo, B. L., Reuning, L., Takayanagi, H., et al. (2021). Intensified organic carbon burial on the Australian shelf after the Middle Pleistocene transition. Quaternary Science Reviews, 262, 106965. https://doi.org/10.1016/j.quascirev.2021.106965

Christensen, B. A., De Vleeschouwer, D., Henderiks, J., Groeneveld, J., Auer, G., Drury, A. J., et al. (2021). Late Miocene Onset of Tasman Leakage and Southern Hemisphere Supergyre Ushers in Near-Modern Circulation. Geophysical Research Letters, 48(18), e2021GL095036. https://doi.org/10.1029/2021GL095036

---

## Author Response (AR2)

Dear editor,

Please find our reply to the editor report and reviewers below.

Dear Suning Hou and co-authors,

I am happy to accept your paper subject to technical corrections.

Thank you very much for your help and acceptance of our manuscript. We also would like to thank both reviewers once more for their efforts and understanding.

While both reviews still would have preferred to see some more details discussion on the Pleistocene variability, and I also think it would have been a valuable addition, they also accept your argument that this will be the focus in a separate paper. I will strongly encourage you to follow up on this for your next paper.

Indeed, we also find the Pleistocene interval interesting and deserve a discussion in greater details. However, this manuscript is really about the long-term SST gradient change in the Southern Ocean, relevant to the Neogene climatic transitions and ocean heat transport. We will definitely draft a separate paper on the Pleistocene part.

Reviewer 1 add some suggestions for clarifications that I would like you to address before final acceptance.

We have replaced 'SST$_{TEX}$' with 'SST$_{TEX86}$', 'SST$_{UK37}$' with 'SST$_{UK'37}$'. We also use subscript numbers in C$_{37}$ and C$_{38}$ alkenones throughout the manuscript.

Best regards,
Bjørg Risebrobakken
Editor, Climate of the Past

Best regards,
Suning Hou and on behalf of co-authors